# In primary airway epithelial cells, the unjamming transition is distinct from the epithelial-to-mesenchymal transition

Jennifer A. Mitchel[1,4], Amit Das[2,4], Michael J. O'Sullivan[1], Ian T. Stancil[1], Stephen J. DeCamp [1], Stephan Koehler[1], Oscar H. Ocaña [3], James P. Butler[1], Jeffrey J. Fredberg[1], M. Angela Nieto [3], Dapeng Bi[2] & Jin-Ah Park [1✉]

The epithelial-to-mesenchymal transition (EMT) and the unjamming transition (UJT) each comprises a gateway to cellular migration, plasticity and remodeling, but the extent to which these core programs are distinct, overlapping, or identical has remained undefined. Here, we triggered partial EMT (pEMT) or UJT in differentiated primary human bronchial epithelial cells. After triggering UJT, cell-cell junctions, apico-basal polarity, and barrier function remain intact, cells elongate and align into cooperative migratory packs, and mesenchymal markers of EMT remain unapparent. After triggering pEMT these and other metrics of UJT versus pEMT diverge. A computational model attributes effects of pEMT mainly to diminished junctional tension but attributes those of UJT mainly to augmented cellular propulsion. Through the actions of UJT and pEMT working independently, sequentially, or interactively, those tissues that are subject to development, injury, or disease become endowed with rich mechanisms for cellular migration, plasticity, self-repair, and regeneration.

[1] Harvard T.H. Chan School of Public Health, Boston, MA, USA. [2] Department of Physics, Northeastern University, Boston, MA, USA. [3] Instituto de Neurociencias (CSIC-UMH), Alicante, Spain. [4] These authors contributed equally: Jennifer A. Mitchel, Amit Das. ✉email: jpark@hsph.harvard.edu

Every organ surface and body cavity is lined by a confluent collective of epithelial cells. In homeostatic circumstances the epithelial collective remains effectively solid-like and sedentary. But during morphogenesis, remodeling or repair, as well as during malignant invasion or metastasis, the epithelial collective becomes fluid-like and migratory[1–4]. This conversion from sedentary to migratory behavior has traditionally been understood as a manifestation of the epithelial-to-mesenchymal transition (EMT) or the partial EMT (pEMT)[5–8]. Since its discovery in 1982, EMT has been intensively studied and well-characterized[6,9,10]. EMT is marked by progressive loss of epithelial character, including disrupted apico-basal polarity, disassembled cell–cell junctions, and impaired epithelial layer integrity and barrier function. This loss of epithelial character is accompanied by progressive gain of mesenchymal character, including gain of front–back polarity, activation of EMT-inducing transcription factors, and expression of mesenchymal markers[11,12]. In this process each epithelial cell tends to free itself from adhesions to immediate neighbors, and thereby can acquire migratory capacity and invasiveness. It has been suggested that the epithelial–mesenchymal axis is flanked at its extremes by unequivocal epithelial versus mesenchymal phenotypes separated by a continuous spectrum of hybrid epithelial/mesenchymal (E/M) or partial pEMT phenotypes[5,8,13–15]. Although such a one-dimensional spectrum of states has been regarded by some as being overly simplistic[5,16], it is widely agreed that pEMT allows cell migration without full mesenchymal individualization[17–20]. During pEMT, cells coordinate with their neighbors through intermediate degrees of junctional integrity coupled with partial loss of apical–basal polarity and acquisition of graded degrees of front–back polarity and migratory capacity[5,8]. Moreover, EMT/pEMT is associated with the cells of highly aggressive tumors, endows cancer cells with stemness and resistance to cytotoxic anticancer drugs, and may be required in the fibrotic response[21,22]. In development[23–26], wound healing[27,28], fibrosis[29], and cancer[30–34], EMT/pEMT has provided a well-accepted framework for understanding collective migration, and in many contexts has been argued to be necessary[2,17,22,28,35,36].

In certain contexts, however, this conversion from sedentary to migratory behavior has been attributed to the recently discovered unjamming transition (UJT), in which epithelial cells migrate collectively and cooperatively[37–40]. By contrast with EMT, UJT in epithelial tissues is recently discovered and remains poorly understood[37–50]. During UJT the epithelial collective transitions from a jammed phase wherein cells remain virtually locked in place, as if the cellular collective were frozen and solid-like, toward an unjammed phase wherein cells often migrate in cooperative multicellular packs and swirls reminiscent of fluid flow. In both the solid-like jammed phase and the fluid-like unjammed phase, the epithelial collective retains an amorphous disordered structure. In the jammed phase, the motion of each individual cell tends to be caged by its nearest neighbors. But as the system progressively unjams and transitions to a fluid-like phase local rearrangements amongst neighboring cells become increasingly possible and tend to be cooperative, intermittent, and heterogeneous[46,51–54]. While poorly understood, cellular jamming and unjamming have been identified in epithelial systems in vitro[37,39,43,46,48,49,54–56], in developmental systems in vivo[39,42,50,57–59], and have been linked to the pathobiology of asthma[37–39] and cancer[45,47,60,61].

Despite strong evidence implicating both pEMT and UJT in the solid–fluid transition of a cellular collective and the resulting collective migration of cells of epithelial origin[35,37–39], the relationship between these transitions remains undefined[62]. For example, it is unclear if UJT necessarily entails elements of the pEMT program. The converse is also in question. As such, we do not yet know if the structural, dynamical, and molecular features of these solid–fluid transitions might be identical, overlapping, or

entirely distinct. To discriminate among these possibilities, we examine mature, well-differentiated primary human bronchial epithelial (HBE) cells grown in air–liquid interface (ALI) culture; this model system is known to recapitulate the cellular constituency and architecture of intact human airway epithelia[63–65].

Here we show that UJT in this system is sufficient to account for vigorous epithelial layer migration in the absence of pEMT. Using the confluent layer of HBE cells, we trigger UJT by exposing the sedentary layer to a mechanical stress that has been tightly linked to aberrant remodeling of the asthmatic airway[37–39,66]. Cells thereafter migrate cooperatively, align into packs locally, and elongate systematically. Nevertheless, cell–cell junctions, apico-basal polarity, and barrier function remain intact in response, and mesenchymal markers remain unapparent. As such, pEMT is not evident. When we trigger pEMT and associated cellular migration by exposing the sedentary layer to TGF-β1, which is known to induce pEMT[22,67], metrics of UJT versus pEMT diverge. To account for these striking physical observations a new computational model attributes the effects of pEMT mainly to diminished junctional tension but attributes those of UJT mainly to augmented cellular propulsion. Together, these findings establish that UJT is sufficient to account for vigorous epithelial layer migration even in the absence of pEMT. Distinct gateways to cellular migration therefore become apparent—UJT as it might apply to migration of epithelial sheets on a collective basis, and EMT/pEMT as it might apply to migration of mesenchymal cells on either a solitary or a collective basis.

## Results

**Cellular dynamics and structure: UJT versus pEMT diverge.** To induce UJT we exposed the cell layer to apical-to-basal mechanical compression of 2.9 kPa (30 cm $H_2O$) for 3 h. This level of compression was chosen for three reasons. First, this level of compressive stress mimics that experienced by the epithelial layer during asthmatic bronchoconstriction[66,68–73]. Second, based upon simple physical arguments this level of compressive stress is readily generalizable to other situations For example, the bulk compressive stiffness modulus of the cell is quite large—on the order of $10^7$–$10^8$ Pa— whereas traction stresses are typically on the order of 100 Pa, intercellular stresses are on the order of 1000 Pa, cellular Young's moduli vary from 100 Pa to 10 kPa depending on cell type[74], and the cortical shear stiffness is on the order of 1000 Pa[37,75–77]. As such, in the vicinity of the lateral intercellular space a compressive stress in the kPa range has been shown to cause localized cellular strains on the order of 0.1, which in turn have been shown to trigger robust intracellular signaling[66]. More generally, forces applied locally or generated endogenously are known to initiate biochemical signaling, cellular deformation and migration[78–80]. Finally, we have established in previous reports that pathologic responses increase with increasing compressive stress but exhibit a maximum response at 20–30 cm $H_2O$. These pathologic responses include asthmatic airway remodeling[81–90] and a robust UJT[37–39].

To induce pEMT we exposed the cell layer to TGF-β1 (10 ng/ml), a well-known EMT-inducing agent[22,67]. Although pEMT has been defined inconsistently in the literature, this dose has been established to induce pEMT in HBE cells in ALI culture[67], a result borne out by our experiments and in line with the criteria defined in a recent consensus statement on EMT[12]. That consensus emphasizes that EMT status cannot be assessed solely on the basis of a single or even a small number of molecular markers. Rather, it asserts that primary criteria for defining EMT status must include morphological and functional cellular properties in conjunction with molecular markers. Accordingly, to assess EMT status we have examined expression and localization of mesenchymal markers (N-cadherin, fibronectin-EDA, vimentin, ZEB-1, Snail1) and epithelial

markers (E-cadherin and ZO-1) together with functional assays (cell migration and layer permeability) and cellular structure (F-actin organization and cell shape). As a time-matched positive control for pEMT we used the cell layer subjected to graded concentrations of TGF-β1 (see "Methods" section). Following exposure to either compressive stress or TGF-β1 in our experimental setup, signatures of UJT and pEMT were evaluated at 24, 48, and 72 h (Supplementary Fig. 1).

In a sedentary confluent epithelial layer, initiation of either UJT or pEMT results in collective migration[8,18,35,37–39]. While the precise dynamic and structural characteristics of the HBE layer undergoing pEMT have not been previously explored, UJT is known to be marked by the onset of stochastic but cooperative

migratory dynamics together with systematic elongation of cell shapes[37–39,49,91,92].

*Dynamics*: We quantified migratory dynamics using average cell speed and effective diffusivity ($D_{eff}$)[37,92]. Control HBE cells were essentially stationary, as if frozen in place, exhibiting only occasional small local motions which were insufficient for cells to uncage or perform local rearrangements with their immediate neighbors (Fig. 1a–c). We refer to this as kinetic arrest or, equivalently, the jammed phase. Following exposure to mechanical compression, however, these cells underwent UJT and became migratory[37–39], with both average speed and effective diffusivity increasing substantially over time and maintained to at least 72 h following compression (Fig. 1a–c, Supplementary Table 1). Following

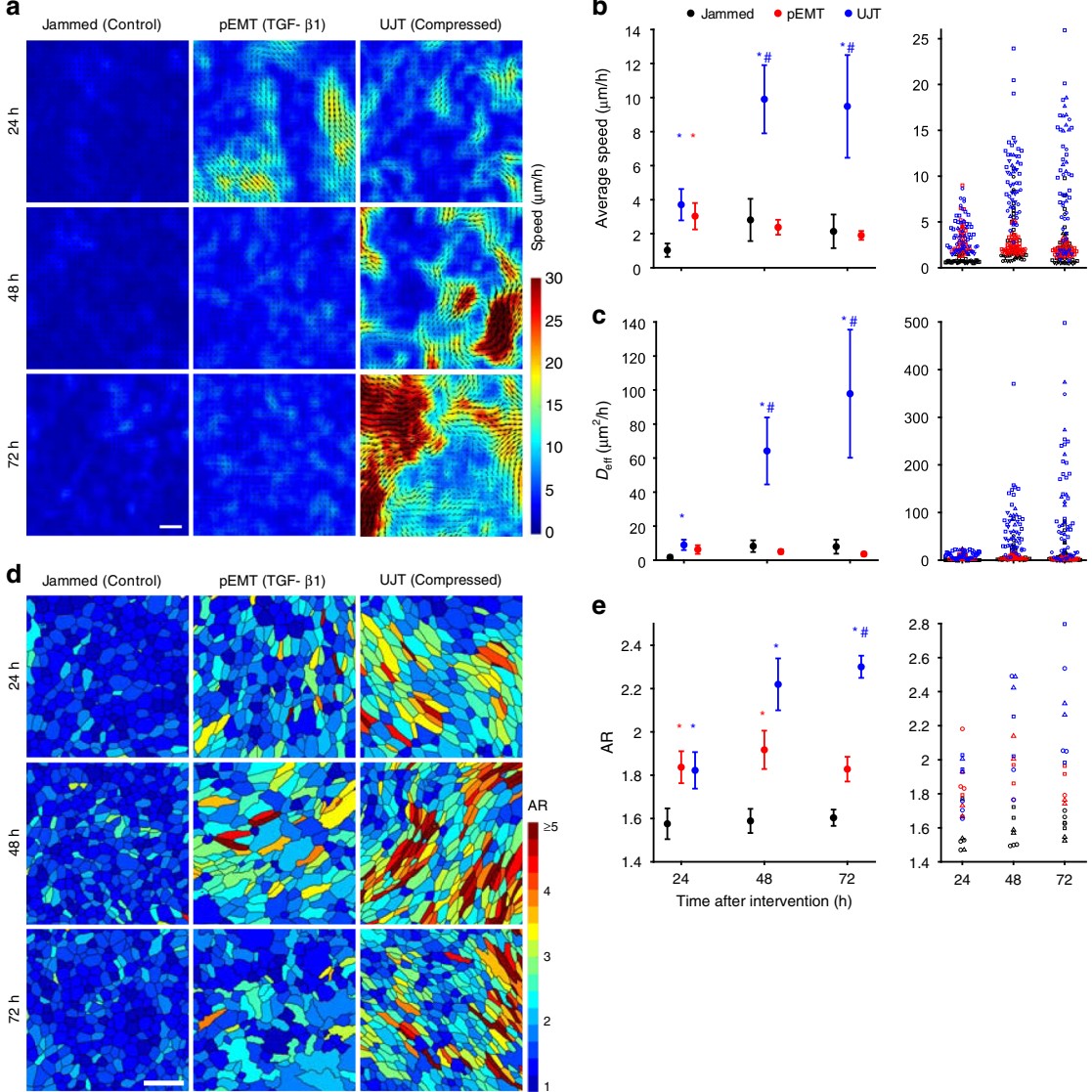

**Fig. 1 In both dynamics and structure, pEMT and UJT diverge over time.** In well-differentiated HBE cells, pEMT was induced by TGF-β1 (10 ng/ml) and UJT was induced by compression[37,66,84,89]. Representative speed maps (obtained using optical flow, see "Methods" section; Scale bar in **a** is 100 μm) **a**, average speed **b**, and effective diffusivity **c** for jammed, pEMT and UJT at 24, 48, and 72 h. **b**, **c** Left panel: mean ± SEM for $n = 4$ donors; right panel: measurements for each field of view across donors, with symbol shape denoting each donor. Jammed cells remain essentially stationary over all times. Cells undergoing pEMT migrate moderately at 24 h but return to baseline by 48 h. Cells undergoing UJT display progressively increasing migratory speed and diffusivity over time. Individual cells color-coded by aspect ratio (AR) **d** and quantified by mean AR (**e**) for jammed, pEMT, and UJT at 24, 48, and 72 h. Scale bar in **d** is 50 μm. **e** Left panel: mean ± SEM for $n = 3$ donors; right panel: measurements for each field of view across donors, with shape denoting each donor. Elevated cellular AR represents a structural signature of the unjamming transition[37,39]. *$p < 0.05$, vs. control; #$p < 0.05$, UJT vs. pEMT, color-coded according to which sample is referenced, one-way ANOVA followed by post-hoc multiple comparisons with Bonferroni correction. Mean (SEM) and full datasets available in Supplementary Table 1 and online Source Data file, respectively.

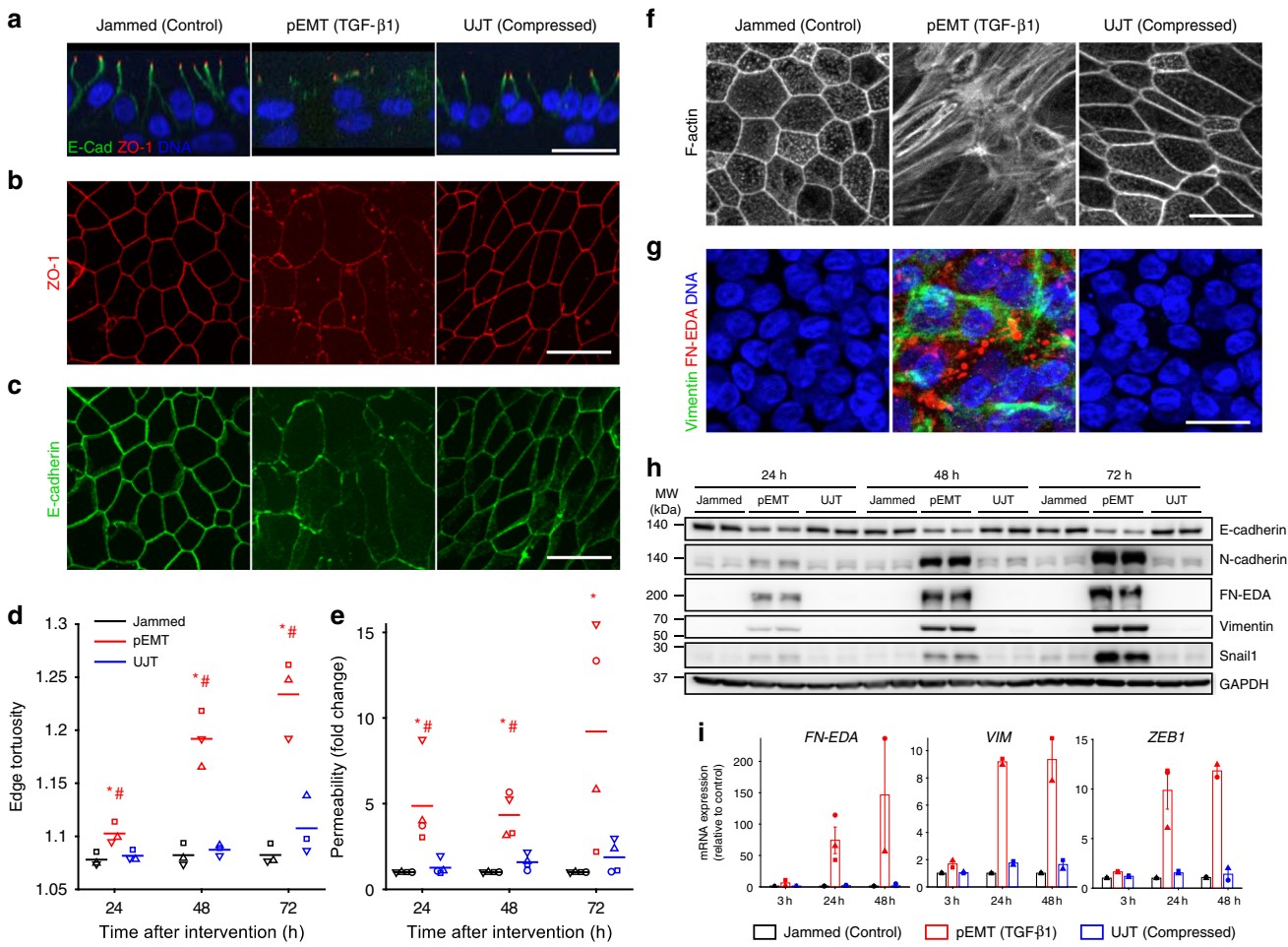

**Fig. 2 UJT occurs without evidence of pEMT.** Representative immunofluorescence (IF) images (**a–c**, **f**, **g**) at 48 h after stimulus for jammed (control), pEMT (TGF-β1-treated), and UJT (compressed) layers. Scale bars in all images are 20 μm. See also Supplementary Figs. 1 and 2. **a** In both jammed and UJT layers, ZO-1 (red) localizes at apical tight junctions, while E-cadherin (green) localizes at lateral adherens junctions, consistent with the epithelial phenotype; DNA is shown in blue. In pEMT layers, both ZO-1 and E-cadherin delocalize from apical and lateral junctions, consistent with the mesenchymal phenotype. **b**, **c** In both jammed and UJT layers, apical tight junctions (ZO-1, **b**) and lateral adherens junctions (E-cadherin, **c**) remain intact, while in pEMT layers the cell edges comprised of these junctions become disrupted. **d** During UJT, cells elongate while maintaining straight cell edges, while during pEMT, cell–cell edges become progressively tortuous ($n = 3$ donors, independent experiments). **e** Layer permeability measured by the dextran-FITC (40 kDa) flux assay significantly increases during pEMT but remains almost unchanged during UJT ($n = 4$ donors, independent experiments). *$p < 0.05$, vs. control; #$p < 0.05$, pEMT vs. UJT, color-coded according to which sample is referenced, one-way ANOVA followed by post-hoc multiple comparisons test with Bonferroni correction **d**, **e**. **f** During pEMT, cortical actin becomes disrupted while apical stress fibers emerge, indicating loss of epithelial character and gain of mesenchymal character. During UJT, cells maintain intact cortical F-actin; aside from elongated cell shape, cortical actin in jammed versus UJT is indistinguishable. **g** IF images stained for mesenchymal marker proteins: cellular fibronectin (FN-EDA, red) and vimentin (green). FN-EDA and vimentin appear during pEMT but not UJT. Vimentin appears as basally located fibers, while FN-EDA appears as cytoplasmic globules. **h** During pEMT, E-cadherin protein levels progressively decrease while mesenchymal markers, N-cadherin, FN-EDA, and vimentin, and the EMT-inducing transcription factor (TF) Snail1, progressively increase. During UJT, these protein levels remained unchanged. Western blot quantification shown in Supplementary Figs. 2, 3, and uncropped versions of the blots are available in the accompanying online Source Data file. **i** During pEMT, mRNA expression of mesenchymal markers, shown as fold-change relative control, *VIM* and *FN-EDA* and the EMT-inducing TF *ZEB1*, are significantly elevated, but during UJT remain unchanged. Shown: mean ± SEM for $n = 3$ donors with overlaid data points showing fold-change for each donor.

exposure to TGF-β1, jammed cells underwent pEMT[22,67], as documented in greater detail below (Fig. 2, Supplementary Figs. 2 and 3). Up to 24 h later these cells migrated with average speeds comparable to cells following compression (Fig. 1a–c). However, as pEMT progressed beyond 24 h cellular motions slowed to the baseline levels, indicating return to kinetic arrest and a jammed phase (Fig. 1a–c, Supplementary Table 1).

*Structure*: We segmented cells from images of cell boundaries labeled for E-cadherin or ZO-1 (Supplementary Fig. 2) and quantified cell shapes using the cellular aspect ratio (AR), calculated as the ratio of the major axis to the minor axis of

the cellular moment of inertia[39]. Control HBE cells exhibited a cobblestone, rounded, and relatively uniform appearance with $\overline{AR} = 1.6$ (Fig. 1d, e). Following exposure to compression, however, cell shapes became more elongated and variable, with progressive growth of $\overline{AR}$ to 2.3 at 72 h (Fig. 1d, e, Supplementary Table 1). Following exposure to TGF-β1, cells elongated to $\overline{AR} = 1.8$ at 24 h but plateaued thereafter (Fig. 1d, e, Supplementary Table 1). As discussed below, the boundaries of cells treated with TGF-β1 also exhibited increased edge tortuosity (Fig. 2d).

After compression, both cell migration and elongation grew over time (Fig. 1). In agreement with previously published work,

these data indicate that the control unperturbed cells exhibited dynamic and structural signatures of a jammed epithelium, while the compressed cells exhibited dynamics and structural signatures of an unjammed epithelium[37–39]. After TGF-β1 treatment, cell migration and elongation initially increased but migration thereafter tapered off and cell shapes remained unchanged. By these dynamic and structural metrics, UJT and pEMT were indistinguishable at 24 h but subsequently diverged. Overall, UJT and pEMT showed distinct profiles of cell migration and shape.

**During UJT, epithelial character persists**. We next investigated the extent to which molecular signatures of pEMT and UJT were distinct or overlapping. Control cells exhibited prominent tight junctions marked by apical ZO-1 and adherens junctions marked by lateral E-cadherin (Fig. 2a–c, Supplementary Fig. 2a–c). ZO-1 and E-cadherin appeared as ring-like structures, demarcating continuous cell boundaries and forming cell–cell junctions (Fig. 2b, c). Cell boundaries were relatively straight (Fig. 2d), suggesting that cell–cell junctions were under the influence of mechanical line tension[93,94]. Further, cells exhibited cortical F-actin rings, which are a hallmark of mature epithelium (Fig. 2f, Supplementary Fig. 3a), and exhibited undetectable or very low expression of mesenchymal markers (Fig. 2g–i, Supplementary Fig. 3b, c). In well-differentiated, mature pseudostratified HBE cells which are jammed, these data serve as a positive control for fully epithelial character.

Exposure to TGF-β1 (10 ng/ml) disrupted epithelial architecture and led to acquisition of mesenchymal character (Fig. 2, Supplementary Figs. 2 and 3). Importantly, the transition through pEMT to full EMT strongly varies depending on both the degree and the duration of the EMT-initiating signal. Induction of full EMT of the well-differentiated HBE layer required extended exposure to TGF-β1 (Supplementary Fig. 4), but here we focus on pEMT. As expected, in response to TGF-β1 both apico-basal polarity and tight and adherens junctions, as marked by ZO-1 and E-cadherin, became progressively disrupted (Fig. 2a–c, Supplementary Fig. 2a–c), and the level of E-cadherin protein decreased (Fig. 2h, Supplementary Fig. 2d). Remaining cell–cell junctions stained for E-cadherin developed increased tortuosity (i.e., the ratio of edge contour length to edge end-to-end distance) suggestive of a reduction in line tension (Fig. 2d)[93,94]. Furthermore, to confirm disruption of barrier function, we measured barrier permeability using dextran-FITC (40 kDa)[81] and observed a substantial increase (Fig. 2e). Cells progressively lost their cortical actin rings while acquiring abundant apical and medial F-actin fibers, a phenotypical feature of mesenchymal cells[95] (Fig. 2f, Supplementary Fig. 3a). Cells also acquired increased expression of EMT-inducing transcription factors including Zeb1 and Snail1, and mesenchymal markers including N-cadherin, vimentin, and fibronectin (EDA isoform) (Fig. 2g–i, Supplementary Fig. 3b, c). Increased expression of these mesenchymal markers occurred simultaneously with disruption of epithelial junctions, thus indicating a clear manifestation of a hybrid E/M phenotype and pEMT. In HBE cells undergoing pEMT, these data serve as a positive control for loss of epithelial character and gain of mesenchymal character.

Exposure to compression (30 cm H$_2$O), by contrast, impacted neither apico-basal polarity nor junctional integrity, as indicated by the apical localization of ZO-1 and lateral localization of E-cadherin (Fig. 2a, Supplementary Fig. 3a). These junctions were continuous (Fig. 2b, c, Supplementary Fig. 2b, c) and nearly straight, thus indicating that during UJT the junctional tension was largely maintained (Fig. 2d). Unlike during pEMT, during UJT the overall level of E-cadherin protein remained unaffected (Fig. 2h, Supplementary Fig. 2d). During UJT cells maintained an apical cortical F-actin ring (Fig. 2f, Supplementary Fig. 3a). While

barrier function was compromised during pEMT, it remained intact during UJT (Fig. 2e). By contrast to cells during pEMT, cells during UJT did not gain a detectable mesenchymal molecular signature (Fig. 2g–i, Supplementary Fig. 3). These data show that epithelial cells undergoing UJT, in contrast to pEMT, maintained fully epithelial character and did not gain mesenchymal character. UJT is therefore distinct from EMT.

Unlike those underlying EMT[12], molecular mechanisms of UJT are largely unexplored. ERK signaling has recently been shown to be involved in UJT in a model of breast cancer[60] and waves of ERK activation regulate collective epithelial migration during wound healing in MDCK monolayers[96,97]. During the compression-induced UJT, a variety of transcriptional and intracellular signaling pathways are activated, including EGFR, PKC, and ERK pathways[66,81,83,87,88,98]. Indeed, we found that blocking ERK activity using the pharmacological inhibitor U0126 (10 μM)[81,99] attenuated compression-induced cellular motility, thus confirming that ERK signaling is required for compression-induced UJT (Supplementary Fig. 5).

**During UJT, cellular cooperativity emerges**. To further discriminate between pEMT and UJT, we next focused on cooperativity of cell shape orientations and migratory dynamics. Because of immediate cell–cell contact in a confluent collective, changes of shape or position of one cell necessarily impacts shapes and positions of neighboring cells; cooperativity amongst neighboring cells is therefore a hallmark of jamming[44,45,52,100,101]. We measured cooperativity in two ways. First, we used segmented cell images to measure cell shapes and shape cooperativity that defined structural packs (Fig. 3a). We identified those cells in the collective that shared similar shape-orientation and then used a community-finding algorithm to identify contiguous orientation clusters (see "Methods" section, Fig. 3a). In both jammed and pEMT layers, cellular collectives formed orientational packs that contained on the order of 5–10 cells and remained constant over time (Fig. 3c). After UJT, by contrast, collectives formed orientational packs that contained 45 ± 22 cells at 24 h and grew to 237 ± 45 cells by 72 h (mean ± SEM, Fig. 3c, Supplementary Table 1).

Second, we used cellular trajectories to measure dynamic cooperativity that defined migratory packs (Fig. 3b). Using optical flow over cell-sized neighborhoods[102], cellular trajectories were constructed by integration. We then used the same community-finding algorithm as above, but here applied to trajectory orientations rather than cell shape orientations (see "Methods" section, Fig. 3b). As a measure of effective pack diameter we used $(4a/\pi)^{1/2}$, where $a$ is pack area. In jammed layers, cellular collectives exhibited small dynamic packs spanning 76 ± 31 μm and containing ~11 ± 7 cells (see "Methods" section, Supplementary Table 1). Interestingly, during pEMT, cells initially moved in dynamic packs spanning 223 ± 67 μm containing ~71 ± 29 cells at 24 h, but these packs disappeared over a time-course matching the disruption of the tight and adherens junctions (Figs. 2b, c and 3b, d). By contrast, during UJT cellular collectives initially exhibited relatively smaller dynamic packs spanning 115 ± 36 μm containing ~19 ± 9 cells at 24 h, but grew to packs spanning 328 ± 74 μm containing ~139 ± 55 cells at 72 h (Fig. 3b, d, Supplementary Table 1).

To determine cellular cooperativity, we employed independent metrics for cellular structure and migratory dynamics. During UJT, structural orientation packs rose monotonically from 24 to 72 h, whereas dynamic orientation packs leveled off from 48 to 72 h (Fig. 3). Despite this unexplained discordance, our data indicate that after UJT, but not after pEMT, structure and dynamics became increasingly cooperative. These observations (Figs. 2, 3, Supplementary Figs. 2 and 3), taken together, indicate

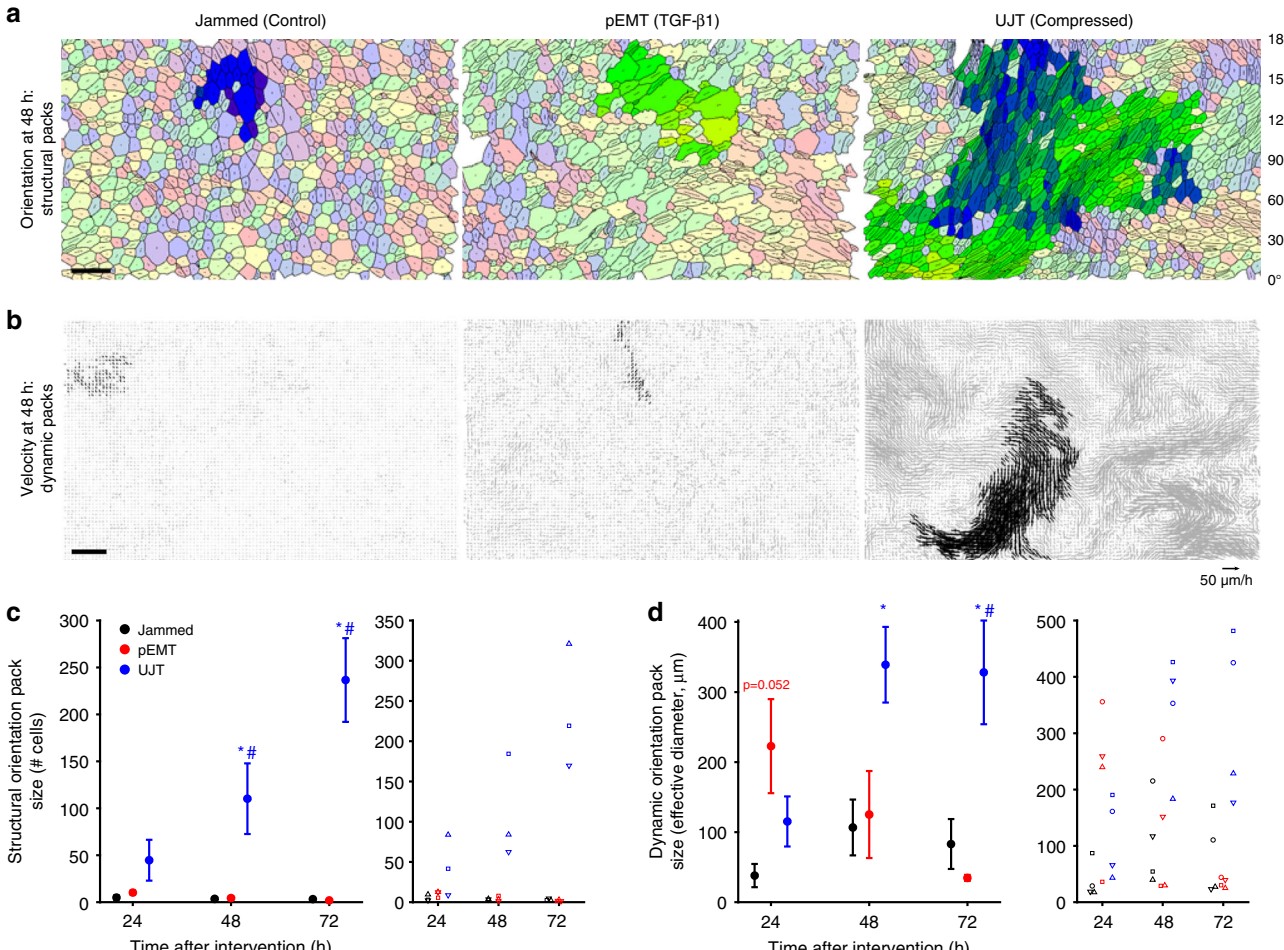

**Fig. 3 During UJT but not pEMT, cellular alignment and cellular migration organize into cooperative packs. a** Using a community-finding algorithm (see "Methods" section), cellular orientations were seen to align into orientational packs. Each cell is shown with an orientation director whose length is proportional to AR. Largest orientational packs highlighted in bold colors. Scale bar is 50 μm. **c** Cells did not exhibit large structural packs when jammed or during pEMT at any time point. ($n = 3$ donors from independent experiments). During UJT, by contrast, cells aligned into packs whose median size progressively grew in cell number from 24 to 72 h. Left panel: mean ± SEM for $n = 3$ donors; right panel: measurements for each donor, delineated by symbol shape. **b** Using velocity fields obtained by optical flow and the same community-finding algorithm, cellular motions were seen to organize into oriented migratory packs. Velocity vectors are shown for a 2-h period, with members of the largest dynamic pack highlighted in black. Scale bar is 100 μm. **d** In jammed layers dynamic packs remained relatively small at all time points ($n = 4$ donors from independent experiments). During pEMT dynamic packs were substantially larger at 24 h but returned to baseline by 48 h. During UJT, by contrast, dynamic packs grew dramatically in size and remained elevated out to 72 h. Left panel: mean ± SEM for $n = 4$ donors; right panel: measurements for each donor, delineated by symbol shape. See also Supplementary Table 1 and online Source Data file. *$p < 0.05$, vs. control; #$p < 0.05$, UJT vs. pEMT, color-coded according to which sample is referenced; one-way ANOVA followed by post-hoc multiple comparisons with Bonferroni correction.

that coordinated cellular movement during UJT occurred in conjunction with maintenance of epithelial morphology and barrier function (Table 1). These data are consistent with an essential role for intact junctions in cellular cooperation[103–108], but are the first to show emergence of coordinated cellular migration in a fully confluent epithelium with no evidence of mixed E/M characteristics or pEMT.

**pEMT versus UJT: discriminating among fluid-like phases.** Results above identify two distinct migratory mechanisms, one arising from pEMT and the other from UJT. To better understand underlying mechanical factors that differentiate pEMT versus UJT, we extend previous computational analyses based on so-called vertex models[37,39,91,92,109–112]. This extended model is described in detail in the Supplementary Methods and is referred to here as the dynamic vertex model (DVM). In the DVM each cell within the confluent epithelial layer adjusts its position and its

shape so as to minimize local mechanical energy. This energy, in turn, derives from three main contributions: deformability of the cytoplasm and associated changes of cell area; contractility of the apical actin ring and associated changes of its perimeter; and homotypic binding of cell–cell adhesion molecules, such as cadherins, together with extensibility of attendant contractile elements and associated changes in cell perimeter[37,109,110]. These structures and associated energies, taken together, lead to a preferred cell perimeter, $p_0$, and determine the tension borne along the cell–cell junction, here called edge tension[37,109,110]. Importantly, contributions of cortical contraction and cell–cell adhesion to system energy are of opposite signs and are therefore seen to be in competition[113]; cortical contraction favors a shorter cell perimeter whereas cell–cell adhesion favors a longer cell perimeter. Equivalently, decreasing cortical contraction causes edge tension to decrease whereas decreasing cell–cell adhesion causes edge tension to increase. As elaborated in the Supplementary Methods,

**Table 1 Across dynamic, structural, and molecular characteristics, pEMT and UJT are distinct.**

| Evidence | pEMT | UJT |
|---|---|---|
| Motility | ↑ | ↑↑↑ |
| Cell elongation | ↑ | ↑↑↑ |
| Junctional Integrity | ↓ | Intact |
| Layer integrity | Disrupted | Intact |
| Apical/basal polarity | Lost | Intact |
| Epithelial markers | ↓ | Intact |
| Mesenchymal markers | ↑ | Not detected |
| Structural packs | No | Yes; increased over time |
| Cooperative motion | Initial trend, then lost | Yes; increased over time |

Across all characteristics, pEMT and UJT diverge. Trends are reported for pEMT and UJT, relative to the jammed condition. These findings establish that UJT is sufficient to account for vigorous epithelial layer migration even in the absence of pEMT.

DVM departs from previous analyses by allowing cell–cell junctions to become curved and tortuous, much as is observed during pEMT. Edge tortuosity can arise in regions where the effects of edge tension becomes small compared with intracellular pressure differences between adjacent cells.

In the DVM, increasing $p_0$ mimics well progressive disruption of the cell–cell-junction and is thus seen to reflect the known physical effects of pEMT (Fig. 4a). For example, when $p_0$ is small and propulsive forces are small the cell layer remains jammed (panel i). Cells on average assume disordered but compact polygonal shapes[37,109] and cell–cell junctions are straight. But as $p_0$ is progressively increased cell shapes become progressively more elongated and cell edges become increasingly curvilinear and tortuous, as if slackened (panels ii and iii). Indeed, edge tensions progressively decrease (as depicted by intensities of the lines) with a transition near $p_0 = 4.1$, at which point edge tensions approach zero and edge tortuosity begins to rise (Fig. 4b). Loss of edge tension coincides with fluidization of the layer and a small increase in cell speed (inset), at which point the shear modulus[114] and energy barriers vanish (Supplementary Fig. 6a). Importantly, for $p_0$ to increase as cell–cell adhesion diminishes, as necessarily occurs as pEMT progresses, DVM suggests that cortical contraction must diminish even faster. Vanishing edge tension in the fluidized state is consistent with the notion that EMT weakens cell–cell contacts, and junctions therefore become unable to support mechanical forces.

When propulsive forces, $v_0$, are increased while $p_0$ is kept fixed, results mimic well the known physical effects of UJT (Fig. 4c). Cell shapes become progressively elongated but cell edges remain straight (panels iv–vi). Edge tension increases but without an increase in edge tortuosity (Fig. 4d). Simultaneously, the speed of the cell migration increases appreciably (inset). This increase in cell speed coincides with fluidization of the layer, at which point cellular propulsion has become sufficient to overcome energy barriers that impede cellular rearrangements (Supplementary Fig. 6b, c). Thus, DVM predicts a dominant role for propulsion during UJT. Indeed, previous experimental work has linked traction forces, propulsion, and collective epithelial migration[115], which has been further shown to require ERK activation[97,116]. ERK activation, which is required for compression-induced UJT (Supplementary Fig. 5), thus provides a mechanistic link between theory and experiment.

During UJT versus pEMT, the DVM predicts, further, that two different metrics of cell shape diverge (Fig. 5a). The cellular AR emphasizes cellular elongation but deemphasizes tortuosity whereas the shape index, $q$ (perimeter/(area$^{1/2}$)) also depends on elongation but emphasizes tortuosity. Indeed, direct measurements of AR versus $q$ from cells undergoing UJT versus pEMT are consistent with the predicted relationship between AR versus $q$ (Fig. 5a, Supplementary Fig. 6d). As regards cell shapes and their changes, UJT versus pEMT are therefore seen to follow divergent pathways. Together, these results attribute the effects of pEMT mainly to diminished edge tension but attributes those of UJT mainly to augmented cellular propulsion. As such, DVM provides a physical picture that helps to explain how the manifestations of pEMT versus UJT on cell shape and cell migration are distinct.

We then used DVM to better understand emergence of collective behavior during UJT (Fig. 3). In promoting collective behaviors, previous computational approaches have pointed toward the importance of cell motility, cell–cell interactions, persistence, confinement, and heterogeneity[43,117–122]. We wish to emphasize, however, that these approaches often impose on an ad hoc basis a local penalty when any given cell fails to align with its immediate neighbors[49,121–124]. Cooperativity and flocking are therefore built into such theories ab initio, and thus are virtually guaranteed to arise. DVM, by contrast, imposes no such penalty. When they arise in DVM simulations, cooperativity and flocking are therefore spontaneous and emergent. Using DVM we assign to each cell a migratory persistence time, $\tau_P$, which naturally gives rise to a single cell migratory persistence length, in units of average cell diameter, $l_0 = v_0 \tau_P / \Gamma$ where $\Gamma$ is the viscous damping coefficient on each cell (Supplementary Methods)[92].

When persistence is small cooperative packs remain few and small. But as persistence progressively increases prominent cell packs are seen to emerge and grow (Fig. 4e, f). Such predicted dependence of the size of the collective emergent pack upon the persistence of individual cells is explained in terms of how the local energy barrier to cellular rearrangement is overcome. When persistence is small the local energy barrier to rearrangement can be overcome, and associated unjamming and migration can occur, only through the application of local cellular propulsive forces (panel vii). In the DVM these localized cellular rearrangements are stochastic and, therefore, result in random uncoordinated patterns of cellular migration and correspondingly small packs (panel vii). When persistence becomes larger, however, cell displacements become spontaneously coordinated across multiple cell diameters, and propulsive forces tend to become aligned and cooperative. As such, the cellular collective tends to unjam via cooperative pack-based migration rather than localized granular rearrangements (panels viii and ix). Equivalently, in the limit of more persistent yet uncoordinated cellular propulsive forces, low frequency (i.e. low energy) elastic mechanical modes become available to the system as opposed to the high-frequency modes which are associated with localized rearrangements. Since low-frequency elastic modes are spatially extended and collective in nature, larger migratory packs emerge[92,125]. This mode of collective migration has also been theoretically predicted recently by Henkes et al.[126].

As persistence is increased, DVM thus predicts that pack size and cell speed increase in concert (Figs. 4f and 5b). Direct measurements of cell speed versus mean pack size from cells undergoing UJT confirm the predicted relationship (Fig. 5b). Emergence of coordinated cell movement facilitates more efficient migration (Figs. 4e, f and 5b).

To explain how the confluent epithelial collective can transition from a solid-like to a fluid-like phase, experimental and computational results, taken together, thus point to a unified physical picture of two distinct mechanisms. In pEMT, fluidization arises mainly from reduction in junctional integrity and loss of edge tension. In UJT, by contrast, fluidization arises mainly from increased propulsion or persistence. Moreover, UJT is

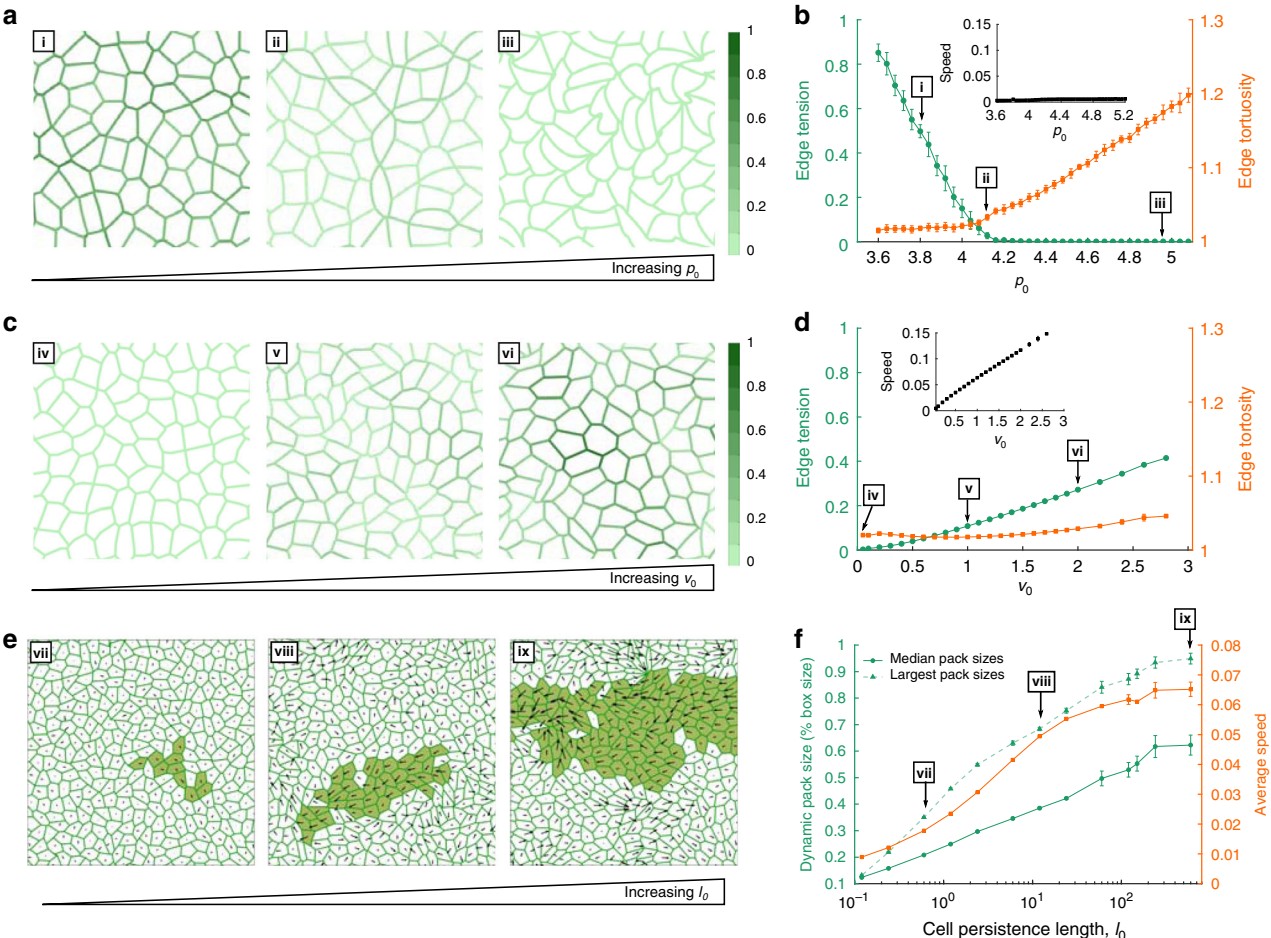

**Fig. 4 Changes in tension, propulsion. and persistence in the dynamic vertex model (DVM) leads to different modes of fluidization in a confluent cell layer. a** When $p_0$ is small and propulsive forces ($v_0$) are sufficiently small, the cell layer is jammed (i). Cells on average assume compact polygonal shapes[37,109], and cell–cell junctions are straight. As $p_0$ progressively increases cell shapes become elongated (ii) and cell edges become increasingly tortuous, as if slackened (iii). Further, as $p_0$ increases, tension in the cell edges decreases (depicted by color intensities). **b** Increasing $p_0$ decreased mean edge tension (green), with a transition near $p_0 = 4.1$, at which point edge tensions dropped to near zero and edge tortuosity began to rise (orange). This loss of edge tension coincides with a solid–fluid transition of the layer, at which point the shear modulus[114] and energy barriers vanish (Supplementary Fig. 4a). **c** If $p_0$ is moderate ($p_0 = 4$) and propulsive force $v_0$ is small the cell layer is immobile (iv). As $v_0$ is progressively increased, however, cell shapes elongate but cell–cell junctions remain straight (v, vi). Further, as $v_0$ increases, edge tensions increase. **d** Increasing $v_0$ increases edge tension (green) but without an increase in edge tortuosity (orange). Simultaneously, the speed of the cell migration increases (inset). This increase in cell speed coincides with fluidization of the layer: $v_0$ becomes sufficient to overcome energy barriers that impede cellular rearrangements. Data presented in **b** and **d** are mean ± SD for $n = 7$–20 independent simulations for each datapoint. **e** Increasing the single cell persistence length $l_0$ (in units of the average cell diameter, see Supplementary Methods) at constant $p_0 = 4$ and $v_0 = 1.2$ leads to the emergence of growing migrating packs (vii–ix)[92,126]. Single cell velocity is represented as a vector; the largest pack is highlighted. **f** The median size of migrating packs (green circles) and corresponding average cell speeds (orange squares) increase in tandem as $l_0$ increases. Data presented in **f** are mean ± SD for $n = 8$–9 independent simulations for each datapoint. The linear sizes of the packs are measured relative to the size of the simulation window (Supplementary Methods). The largest migrating packs are also shown (green triangles), which grow to the size of the simulation window for $l_0 > 100$, highlighting the scalability and robustness of pack formation in DVM.

marked by emergent migratory packs together with a systematic tendency for cellular elongation.

## Discussion

Development, wound repair, and cancer metastasis are fundamental biological processes. In each process cells of epithelial origin are ordinarily sedentary but can become highly migratory. To understand the mechanisms by which an epithelial layer can transition from sedentary to migratory behavior, the primary mechanism in many contexts had been thought to require EMT or pEMT[5,127–130]. During EMT/pEMT cells lose apico-basal polarity and epithelial markers, while they concurrently gain front-to-back polarity and mesenchymal markers. Each cell thereby frees itself from the tethers that bind it to surrounding cells and matrix and assumes a migratory phenotype. In the process, epithelial barrier function becomes compromised. Here by contrast we establish the UJT as a distinct migratory process in which none of these events pertain. Collective epithelial migration can occur through UJT without EMT or pEMT.

EMT/pEMT refers not to a unique biological program but rather to any one of many programs, each with the capacity to confer on epithelial cells an increasingly mesenchymal character[12,127]. In doing so, EMT/pEMT tends to be a focal event wherein some cue stimulates a single cell—or some cell subpopulation—to delaminate from its tissue of origin and thereafter migrate to potentially great distances[26,131]. As such, EMT likely evolved as a mechanism that allows individual epithelial cells or

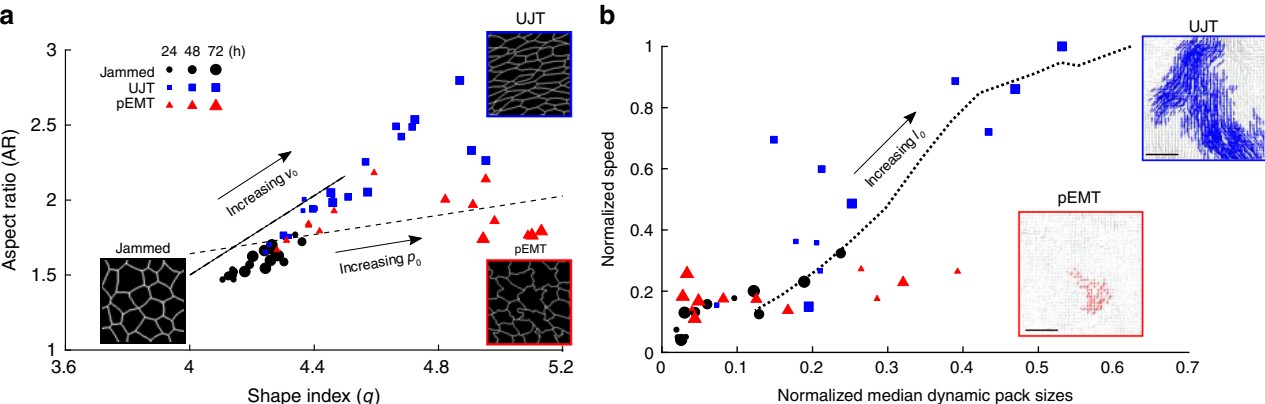

**Fig. 5 In theory and experiment, cell shape and collective dynamics discriminate UJT from pEMT.** The dynamic vertex model (DVM) attributes the effects of pEMT mainly to diminished edge tension but attributes those of UJT mainly to augmented cellular propulsion. **a** DVM predicts that during UJT versus pEMT two different metrics of cell shape diverge; aspect ratio (AR) emphasizes elongation whereas shape index $q$ emphasizes perimeter ($q =$ perimeter/(area$^{1/2}$)). Increasing $p_0$ (- - -) moderately increases AR but substantially increases $q$, resulting in somewhat elongated cells with tortuous edges. By contrast, increasing $v_0$ (-·-·-) substantially increases AR but minimally increases $q$, resulting in elongated cells with straight edges. Measurements of AR and $q$ from cells undergoing UJT (blue squares) or pEMT (red triangles) are consistent with those predictions. Insets show traced cell edges from representative images of cells in each state. During pEMT edge tension decreases as junctional adhesion decreases, and as cells elongate $q$ increases more quickly than AR. Cell–cell junctions become increasingly tortuous and slack. During UJT, by contrast, edge tension increases as cellular propulsion $v_0$ increases, AR and $q$ increase in tandem, and cells elongate. Cell–cell junctions remain straight and taut. Theory and experimental data, taken together, suggest that layer fluidization by means of UJT versus pEMT follow divergent pathways. **b** DVM predicts that median pack size and average cell speed will, as persistence length $l_0$ (···) increases, positively scale together. The time courses of median pack size and speed from jammed cells (black circles) or those undergoing pEMT (red triangles) did not follow this relationship. However, the time evolution of median pack size and speed from cells undergoing UJT (blue squares) are consistent with these predictions. During UJT, cellular cooperativity emerges and cells are able to move quickly in locally coordinated groups. Pack sizes, for both experimental and simulation data, were scaled by the size of the system, while cell speeds were scaled by the global maximum. Insets show representative dynamic packs from cells undergoing pEMT or UJT (scale bar: 100 μm). DVM simulation and experimental data, taken together, thus provide a physical picture of two distinct mechanisms of transition of a cellular collective from a solid-like to a fluid-like state.

cell clusters to separate from neighbors within the cell layer and thereafter invade and migrate through adjacent tissue[132]. Like EMT/pEMT, the UJT is observed in diverse contexts and may encompass a variety of programs[37–39,41–49,87]. But by contrast with EMT/pEMT, UJT comprises an event that is innately collective, wherein some cue stimulates cells constituting an integrated tissue to migrate collectively and cooperatively[133]. Due to the presence of a non-degradable basal transwell-insert, epithelial cells in our system cannot invade, and thus we could not compare invasion phenotypes between UJT and pEMT. Nonetheless, our data are consistent with the hypothesis that UJT might have evolved as a mechanism that allows epithelial rearrangements, migration, remodeling, plasticity, or development within a tissue under the physiological constraint of preserving tissue continuity, integrity, and barrier function.

We establish here in the mature layer of primary HBE cells that UJT does not require pEMT. That finding in turn motivates three new questions. First, UJT has now been observed across diverse biological systems[37–39,41–49], but we do not yet know whether UJT is governed across these diverse systems by unifying biological processes or conserved signaling pathways. Second, although we now know that UJT can occur in the absence of pEMT, it remains unclear if pEMT can occur in the absence of UJT. This question is illustrated, for example, by the case of ventral furrow formation during gastrulation in the embryo of *Drosophila melanogaster*, which requires the actions of EMT transcription factors[134–136]. Prior to full expression of EMT and dissolution of cell–cell junctions in *Drosophila*, embryonic epithelial cells have been shown to unjam; cell shapes elongate and become more variable as cells begin to rearrange and migrate[39]. Supporting that notion, our data in HBE cells point towards a role for UJT in the earliest phase of pEMT; when junctional disruption and expression of EMT transcription factors and mesenchymal markers are

apparent but minimal (24 h; Supplementary Figs. 2 and 3), wherein cells are seen to unjam, elongate, and migrate in large dynamic packs (Figs. 1a, b and 3d). These observations argue neither for nor against the necessity of EMT for progression of metastatic disease[127,129,137,138], but do suggest the possibility of an ancillary mechanism.

In many cases the striking distinction between EMT/pEMT versus UJT as observed here is unlikely to be so clear cut. It has been argued, for example, that EMT-induced intermediate cell states are sufficiently rich in their confounding diversity that they cannot be captured along a linear spectrum of phenotypes flanked at its extremes by purely epithelial versus mesenchymal states[5,16]. In connection with a cellular collective comprising an integrated tissue, observations reported here demonstrate, further, that fluidization and migration of the collective is an even richer process than had been previously appreciated. Mixed epithelial and mesenchymal characteristics, and the interactions between them, are thought to be essential for carcinoma cell invasion and dissemination[13,14,16,20,120], but how UJT might fit into this physical picture remains unclear[139,140]. More broadly, the Human Lung Cell Atlas now points not only to dramatic heterogeneities of airway cells and cell states, but also to strong proximal-to-distal gradients along the airway tree[141]. But we do not yet know how these heterogeneities and their spatial gradients might impact UJT locally, or, conversely, how UJT might impact these gradients. In that light, the third and last question raised by this work is the extent to which EMT/pEMT and UJT might work independently, sequentially, or cooperatively to effect morphogenesis, wound repair, and tissue remodeling, as well as fibrosis, cancer invasion, and metastasis[142,143].

## Methods

**Cell culture**. Primary HBE cells at passage 2 were differentiated in ALI, described below[37,81,83–85]. Primary HBE cells were isolated at Passage 0 at the Marsico Lung

Institute/Cystic Fibrosis Research Center at the University of North Carolina, Chapel Hill. Human lungs unsuitable for transplanataion were obtained under protocol #03-1396 approved by the University of North Carolina at Chapel Hill Biomedical Institutional Review Board. Informed consent was obtained from authorized representatives of all organ donors. Lungs were from non-smokers with no history of chronic lung disease. Demographic information is available for all donors used in our study upon request. Cells were expanded to Passage 2 in our lab, and used for all experiments at Passage 2, as described in the manuscript.

Passage 2 primary HBE cells were plated onto type I collagen (0.05 mg/ml) coated transwell inserts (Corning, 12 mm, 0.4 µm pore, polyester) and maintained in a submerged condition for 4–6 days. Culture media consisted of a 1:1 mixture of DMEM (high glucose, 4.5 g/L) and bronchial epithelial basal medium (BEBM, Lonza) supplemented with bovine pituitary extract (BPE, 52 µg/ml), epidermal growth factor (EGF, 0.5 ng/ml), epinephrine (0.5 µg/ml), hydrocortisone (0.5 µg/ml), insulin (5 µg/ml), triiodothyronine (6.5 ng/ml), transferrin (10 µg/ml), gentamicin (50 µg/ml), amphotericin-B (50 ng/ml), bovine serum albumin (1.5 µg/ml), nystatin (20 units/ml), and retinoic acid (50 nM). Thus, for the entire culture period, HBE cells were maintained in defined, serum-free media[81]. Once the layer became confluent, medium was removed from the apical surface and the ALI condition was initiated. Over 14–17 days in ALI, the cells differentiated and formed a pseudostratified epithelium which recapitulated the cellular architecture and constituency of the intact human airway[39,63,64,86,144]. Prior to the experiments, cells were maintained for 20 h with minimal medium depleted of EGF, BPE, and hydrocortisone. For experiments with time points longer than 24 h, cells were fed with fresh minimal media at 48 h following the initial media change prior to exposure.

Experiments were repeated with primary cells from at least $n = 3–4$ donors in independent experiments. HBE cells were derived from donors with no history of smoking or respiratory disease, as used in our previous studies[37,81,83–85]. Experimental quantifications are shown across all donors and reported $n$ is number of independent donors used.

To initiate pEMT, cells were treated with recombinant human TGF-β1 (10 ng/ml, Cell Signaling Technology)[67]. This dose of TGF-β1 was chosen according to a dose-dependent experiment at 1, 10, and 50 ng/ml, at which EMT is induced in a variety of systems[145–149]. In well-differentiated HBE cells, 10 ng/ml is an effective dose to induce hallmarks of complete EMT at 14 days (Supplementary Fig. 4). Our analysis was performed between 24 and 72 h, while cells exhibited widely accepted signatures of pEMT[5,8]. Based on dose, we found slight variations in the exact levels of epithelial and mesenchymal markers but our conclusions remain unchanged. To initiate UJT, cells were exposed to mechanical compression with an apical-to-basal pressure differential of 30 cm $H_2O$[37,81–85]. Briefly, silicon plugs with an access port were press-fit into the top of each transwell. Access ports were either open to room air for sham controls or connected to 5% $CO_2$ (balanced room air) compressed to 30 cm $H_2O$. Cells were exposed to compressed air for 3 h. Time-matched control cells were set up with vehicle treatment for TGF-β1 and a sham pressure for mechanical compression. For each donor and experiment, time-matched controls, TGF-β1-treated, and compressed conditions were all performed in parallel, with each experiment stopped at the indicated time (Supplementary Fig. 1 for experimental setup).

**Protein and mRNA expression analysis.** We detected protein levels by western blot analysis as described previously[81]. Cell lysates were collected into 150 µl 2× Laemmli buffer with 1 M DTT at 24, 48, or 72 h after initial exposure to stimuli (vehicle/sham, TGF-β1 at 10 ng/ml, or compression at 30 cm $H_2O$). The following antibodies and dilutions were used, with primary antibody diluted in 5% skim milk or 5% BSA according to the manufacturer's instructions: E-cadherin (1:10,000), N-cadherin (1:1000), Snail1 (1:1000), vimentin (1:1000), GAPDH (1:5000), all from Cell Signaling Technology; EDA-fibronectin (1:1000, Sigma). We report fold-changes of normalized protein levels compared either to vehicle control (for E-cadherin) or to TGF-β1—treated at 72 h (for mesenchymal markers) across $n = 3$ donors.

We detected mRNA expression as previously described[84]. Cells were collected from the conditions and donors as described above at 3, 24, or 48 h after the initial exposure to stimuli, and RNA was isolated from cell lysates using the RNeasy Mini Kit (Qiagen) following the manufacturer's instructions. Real-time qRT-PCR was performed using primers listed in Supplementary Table 2, and fold-changes were calculated by the comparative ΔΔCt method[150].

**Immunofluorescence staining.** At 24, 48, or 72 h after initial exposure to stimuli, cells were fixed with either: 4% paraformaldehyde in PBS with calcium and magnesium for 30 min at room temperature; or, 100% methanol at −20 °C for 20 min. Cells were permeabilized with 0.2% Triton X-100 for 15 min and blocked with 1% bovine serum albumin and 10% normal goat serum for 1 h. Cells were stained for F-actin (Alexa fluor 488-Phalloidin, 1:40, 30 min) or for proteins of interest, as follows: E-cadherin (1:200, Cell Signaling Technology), ZO-1 (1:100, Thermo-Fisher), vimentin (1:100, Cell Signaling Technology), cellular fibronectin (Extra Domain A splice variant, denoted FN-EDA, 1:200, EMD Millipore). Cells were counterstained with Hoechst 33342 (1:5000) for nuclei. Following staining, transwell membranes were cut out from the plastic support and mounted on glass slides (Vectashield). Slides were imaged using Zen Blue 2.0 software on a Zeiss Axio

Observer Z1 using an apotome module. Maximum intensity images were generated in ImageJ (v 1.52n). Side view images were reconstructed from a z-stack, while top down images were maximum intensity projections generated ~10 µm of the z-stack.

**Live imaging and dynamic analysis.** To determine cellular dynamics, time-lapse movies were acquired and analyzed. Images were taken every 6 min for 6 h, ending at 24, 48, or 72 h after initial exposure to stimuli. Phase contrast images were acquired using Zen Blue 2.0 software on a Zeiss Axio Observer Z1 with stage incubator (37 °C, 5% $CO_2$). Time-lapse movies were analyzed using custom software written in Matlab (R2019a). Cellular dynamics were determined using an optical flow algorithm. The movies were registered to sub-pixel resolution using a discrete Fourier transform method[151]. Flow fields were calculated from the registered movies using Matlab's OpticalFlowFarneback function (R2019a). Trajectories were seeded from the movie's first frame using a square grid with spacing comparable to the cell size and obtained from forwards-integration of the flow fields; for our field of view there were about 4000 trajectories. The average speed was calculated from the displacement during a two-hour window, and the effective diffusivity was calculated from the slope of the mean square displacement.

**Permeability.** Epithelial barrier function was determined by a dextran-FITC flux assay, as described previously[81]. Directly following time-lapse imaging of HBE cells, 1 mg/ml dextran-FITC (40 kDa; Invitrogen) was added to the apical surface of cells. After 3 h, medium was collected from the basal chamber, and used for measuring fluorescence intensity of FITC. Fluorescence intensity measured in media from stimulated cells is expressed as fold-change relative to that in the media from time-matched control cells.

**Cell shape analysis.** To determine cell shape distributions, we marked cellular boundaries and measured shape characteristics as described below. To mark cellular boundaries, we segmented immunofluorescent cell images using SeedWater Segmenter (v0.5.7.1)[13]. Images used were maximum intensity projections of ZO-1 and E-cadherin at the apical region of the cell layer. Segmented images were used to determine cell boundaries and extract cell shape information, including apical cell area, perimeter, and AR from major and minor axes of an equivalent ellipse. This fitted ellipse has equivalent eigenvalues of the second area moment as of the polygon corresponding to the cell boundaries, as published previously[39]. In addition to cell AR, we computed the cell shape index $q = \mathrm{perimeter}/\sqrt{\mathrm{area}}$. We also extracted individual cell edges and computed the end-to-end distance and the contour distance along the edge to compute the edge tortuosity:

$$\mathrm{Tortuosity} = \frac{\mathrm{contour\ length}}{\mathrm{end-to-end\ length}} \qquad (1)$$

**Structural and dynamic cluster analysis.** Orientation clusters, or packs, were determined from both cell shape orientation and from cell trajectory orientation, using a community-finding algorithm as described below. Cell shape orientation was determined from segmented immunofluorescent cell images, while cell trajectory orientation was determined from dynamic flow fields. Each cell or trajectory possessed orientation $\theta_j$ with respect to a global axis of reference. The method below was developed for cell shape orientation clusters and was then applied to cell trajectories to determine dynamic orientation clusters. The determination of orientation clusters started by initiating a neighbor-count on each cell in a given image. We detected the number of neighbors $m_i$ of the $i$th cell possessed similar orientations within a cutoff $\delta\theta = \pm 10°$. This led to an increase of neighbor-count on each of these neighbor cells of cell $i$ by the number $m_i$. We created the set of these neighbor cells for cell $i$ and repeated this neighbor-finding for each of the other members in the set except cell $i$. We increased the neighbor-counts on all the members by the newly found number of neighbors and updated the set of connected cells. We continued to look for neighbors for all the new members of the set until we were unable to find a neighbor with similar orientation for any new member. This gave us a cluster of structurally connected cells where each of the cells have at least one neighbor with orientation within $\delta\theta$. We called this an orientation-based cluster or a structural pack. We determined the mean pack-size per cell by counting, for each cell, the number of cells in its pack, and averaging. This can be expressed mathematically as follows: if in the $j$th structural pack there are $s_j$ number of cells, and there are $N_c$ cells in an image, the mean pack-size per cell would be $s = (1/N_c)\sum_{j=1}^{N_c} s_j$. A null test for our algorithm was to set $\delta\theta = \pm 90°$ and find that all cells in an image became part of the same connected cluster giving a mean pack-size equal to the number of cells.

We performed the same pack-size analysis on the cellular trajectories obtained from the velocity field determined using optical flow. We applied a uniform speed threshold equal to the mean speed on each image and then a cutoff on the orientations of velocity vectors given by $\delta\theta = 10°$. The rest of the calculation proceeded as above. Once we obtained the number of velocity vectors in each dynamic pack, we converted this to a two-dimensional area corresponding to the size of the pack. We then expressed an effective pack size according to the $(4a/\pi)^{1/2}$, where $a$ is pack area. We also converted this areal pack size into an approximate

number of cells by using the average cell size determined for control cells for four donors, from the shape analysis described above.

**Statistics and reproducibility**. All of the data was analyzed in Matlab using custom scripts. To determine statistical significance, we ran an ANOVA for each data set, comparing across the multiple donors used. This was followed by a post-hoc analysis using a Bonferroni correction, and $p < 0.05$ was considered significant.

All experiments were repeated independently with HBE cells derived from at least three donors with two biological replicates per condition and timepoint. Dynamic measurements (Figs. 1c, d and 3d; Supplementary Fig. 5b) and functional measurements (Fig. 1d) were repeated in $n = 4$ donors while structural measurements (Figs. 1e, 2d, and 3c) were repeated in $n = 3$ donors. Individual data points for each donor are shown. For each biological replicate used to obtain dynamic measurements, corresponding to an individual transwell, timelapse imaging movies were taken from 6 to 18 fields of view, from which ~4000 trajectories were obtained via optical flow, described above. For each biological replicate used to obtain structural measurements, two fields of view were taken, from which ~2000 cells were evaluated.

Protein measurements (Fig. 2h, Supplementary Figs. 2d and 3d) were repeated in $n = 3$ donors from independent experiments with two biological replicates per condition and timepoint. Western blots were loaded in parallel and both run and imaged with identical conditions, and each blot was normalized to its own internal loading control of GAPDH. The blots show representative data which was consistent across the $n = 3$ donors used.

Immunofluorescent images comparing three experimental interventions across three timepoints (Fig. 2a–c, f, g and Supplementary Figs. 2a–c, 3a, b) were not quantified. Included images were representative, and display morphology and localization of epithelial and mesenchymal markers that was consistent across $n = 3$ donors and two biological replicates per donor.

**Reporting summary**. Further information on research design is available in the Nature Research Reporting Summary linked to this article.

## Data availability
Data, both raw and analyzed, that comprise the graphs within this manuscript and other findings of this study are available from the corresponding author upon request. The source data underlying all graphs, along with uncropped western blots, are provided as a Source Data file. Source data are provided with this paper.

## Code availability
The code used to process the data and generate the graphs within this manuscript is available from the corresponding author upon request.

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

## Acknowledgements

The authors thank Jeffrey M. Drazen for his critical feedback. The authors acknowledge the support of the Northeastern University Discovery Cluster and TIER 1 Seed Grant for interdisciplinary research, P01HL120839, R01HL148152, U01CA202123, T32HL007118, P30DK065988, the Parker B. Francis Foundation, American Heart Association (13SDG 14320004), the Spanish Ministry of Science, Research, and Innovation (RTI2018-096501-B-I00).

## Author contributions

J.A.M., M.A.N., J.-A.P. designed experiments; J.A.M., M.J.O., I.T.S., O.H.O. performed experiments; A.D. and D.B. performed dynamic vertex model simulations; J.A.M., A.D., M.J.O., S.K., D.B. analyzed data; J.A.M., A.D., S.J.D., J.P.B., J.J.F., M.A.N., D.B., J.-A.P. interpreted data; J.A.M., A.D., J.J.F., D.B., J.-A.P. wrote the manuscript.

## Competing interests

The authors declare no competing interests.
