## [Peer Review File · Nature Communications]

Reviewers' comments:

Reviewer #2 (Remarks to the Author):

This article tackles the interesting question of the similarities and differences between EMT and UJT. It's an important question and the authors make a case that these two transitions are fundamentally different. They then explain these differences by playing on the various parameters of a vertex model, coming to the conclusion that EMT is more local and dependent on edge tension while UJT is more collective and cell-speed dependent.

The initial question is very clear and well-posed. Differences illustrated in Fig1-3 indeed point to a difference of nature between these two transitions. However, the authors rely on the concept of partial EMT that is real but also ill-defined. If I understand correctly the experiments are performed between 1 day and 3 days after addition of TGF. In this period of time, the cell layer continuously evolves (Supp Figure 3). Have the authors tested other (larger) concentrations of growth factors? Have they tried to perform their experiments at longer times? In other words is it possible that the system would evolve differently depending on the TGF dose and/or time of the culture? The same being true of the mechanical stimulation: why 30 cm H₂O and not another value? I fully agree that these two conditions separately trigger the expected phenotypes. However, since the idea of the present paper is to compare them, the "right" conditions are difficult to identify.

At that point the vertex model-based simulations should be very valuable. And it is in some respect. The observed trends qualitatively reproduce the experimental observations and the interpretation that the authors give although qualitative seem sound and give a very plausible picture of the two transitions. However, it seems to me that it should be possible to go further and to extract real numbers from the simulations. In other words, to somehow "fit" the theoretical behaviors on the actual situation. Why aren't tortuosity, aspect ratio or pack sizes directly compared between the experiments and the theory? For instance, the pack size encompasses typically 10 cells in the theory and is much larger in the UJT experiments. The introduction of the persistence time and its importance would also deserve more explanation. As it is, it appears unrelated to the other parameters and although somehow natural, it is unclear what are the conclusions derived from Figure 4f. In general, the authors should include more elements of the model in the main text. In the last part, the authors mention the processes of invasion through adjacent tissues. This is quite different from the situation studied here and the phrasing should be revised. Finally, I don't understand the last part mentioning the potential role of UJT in the first steps of pEMT. I don't see to what the authors refer to (next to last paragraph).

All in all, the article is beautifully written, it addresses an important point, and it provides some clear answers, even if the above-mentioned points question the black-and-white conclusions. I would therefore favor its publication in Nature Communications providing that the comments above are addressed in the text to make sure the reader can understand the hypotheses and limitations of the analysis.

Reviewer #3 (Remarks to the Author):

The paper entitled "In airway epithelial cells, the unjamming transition is distinct from the epithelial-

to-mesenchymal transition” from Mitchel et al., presents original experimental results related to the transition from stable epithelia to dynamic structures where the cellular organization can remodel. The authors use two different types of perturbation, mechanical pressure and exposure to TGF-beta1, to trigger this transition. Interestingly they found that, although both approaches enable cell migration, the qualitative behavior of the cells was different in these two situations. Using a model, the authors interpret the role of TGF-beta1 as reducing junctional tension and allowing the cells to explore a richer landscape of shapes, facilitating rearrangements. Pressure, however, was seen to trigger significant motion and was interpreted in the model as increased cellular propulsion.

Demonstrating both patterns of behavior with the same experimental system (but using different treatments) is interesting and gives a lot of value to his paper. It brings together different interpretations of the transition from stable epithelia to motile layers. As such, the experimental content of paper is worth publishing, and the model provides useful ideas to interpret the data. There are various aspects that the authors may need to clarify before the paper can be published.

1. The compression protocol needs to be explained more thoroughly. What is being compressed and how? Is it uniaxial compression? What pressure is imposed? How could cells sense this mechanical signal? Why would they move under compression? Apologies if I missed this explanation in the submission.

2. The vertex model extension used in this study illustrates well the point being made. But this model is an incremental improvement on the work of many others over the past 10 years. I'm not saying this as a negative point, but more credit needs to be given to precursors of the ideas developed here, highlighting the novelty where appropriate. The authors discuss in particular on page 8 (bottom) the benefits on their approach based on a persistence time. It should be clear that a number of older studies from the early 2010s have already studied the role of migratory forces as a trigger for a dynamic transition, in some cases in a way that is very similar to the present model. See for instance [doi:10.1098/rsif.2012.0448](https://doi.org/10.1098/rsif.2012.0448) as well as its extension to the case of heterogeneous cells populations, [doi:10.1098/rsos.161007](https://doi.org/10.1098/rsos.161007) . Along the same line, when the authors say in their conclusion that “mixed epithelial and mesenchymal characteristics, and the interactions between them, are thought to be essential for carcinoma cell invasion and dissemination, but how UJT might fit into this physical picture remains unclear”, it is worth pointing that such questions have already been touched upon by people studying collective movements. Although they did not call their transition “UJT”, the underlying models would have very similar physical foundations.

Response to Editor:

We thank both reviewers for their thoughtful comments and helpful suggestions. Indeed, the reviews were immensely helpful, and we are most appreciative. As much as possible we have worked to improve in clarity and coherence. In the revised manuscript and in the responses below we address these critiques and provide new data to address the concerns raised by the reviewers. Substantive changes from the previous draft are shown in blue.

Reviewer #2 (Remarks to the Author):

Comment: This article tackles the interesting question of the similarities and differences between EMT and UJT. It's an important question and the authors make a case that these two transitions are fundamentally different. They then explain these differences by playing on the various parameters of a vertex model, coming to the conclusion that EMT is more local and dependent on edge tension while UJT is more collective and cell-speed dependent. The initial question is very clear and well-posed.

Response: Thank you.

Comment: Differences illustrated in Fig1-3 indeed point to a difference of nature between these two transitions. However, authors rely on the concept of partial EMT that is real but also ill-defined.

Response: This is a terribly important point. Thank you for raising it. Indeed, this very issue is the subject of a new consensus statement that has just appeared and is authored by nearly fifty leaders in the EMT field¹. That consensus statement emphasizes that EMT status cannot be assessed on the basis of one or even a small number of molecular markers. Rather, it asserts that primary criteria for defining EMT status must be a multifaceted combinatorial approach in which changes in cellular morphological and functional properties are considered in conjunction with molecular markers.

That is precisely the approach that we have taken. To assess EMT status we have included expression and localization of mesenchymal markers (N-cadherin, fibronectin-EDA, vimentin, ZEB-1, snail) and epithelial markers (E-cadherin and ZO-1) together with functional markers (cell migration and layer permeability). Using immunofluorescence, western blotting, and qPCR to show both localization and expression of the protein and expression of mRNA together with structural and functional markers, we have rigorously validated presence of pEMT in well-differentiated airway epithelial cells. Our evaluation of all these factors, taken together, offers strong confirmation of an epithelial state and, by means of stimuli described below, the triggering of pEMT. We have revised both the main text and methods to clearly justify the definitions and methods, including the dose and duration that we used to induce pEMT and UJT (lines 84-93 and lines 401-412).

Comment: If I understand correctly the experiments are performed between 1 day and 3 days after addition of TGF. In this period of time, the cell layer continuously evolves (Supp Figure 3). Have the authors tested other (larger) concentrations of growth factors? Have they tried to perform their experiments at longer times? In other words is it possible that the system would evolve differently depending on the TGF dose and/or time of the culture?

Response: Yes! As now described in the methods (lines 401-406), we have tested a range of concentrations (1-50 ng/ml) for TGF- β 1. In the manuscript we show extensive data for 24, 48 and 72 hours of TGF- β 1 treatment (Extended data Fig. 2, 3). We also show data for 4 – 14 days of TGF- β 1 treatment (Extended data Fig. 4). Our findings align well with the literature, which defines pEMT as progressive loss of epithelial character coupled with progressive gain of mesenchymal character, and therefore support our choice of conditions to generate samples which act as a positive control for pEMT. Also in agreement with the literature, our pilot experiments indicated that this transition can be relatively slowed or accelerated by treatment with lower or higher doses of TGF- β 1. Our goal was not to exhaust all possible combinations of dose and timing, but rather to find a set of conditions with unequivocal characteristics of pEMT and use this as a positive control. Accordingly, we chose to use 10 ng/ml TGF- β 1 and look at time points in parallel with the UJT.

Comment: The same being true of the mechanical stimulation: why 30 cm H₂O and not another value?

Response: As now described in the text (lines 84-87), we used apico-to-basal compression not only because this mechanical stimulus is transduced by the cells but also because it mimics the manner in which asthmatic bronchospasm compresses the airway epithelium and thus promotes aberrant airway remodeling that is a cardinal feature of asthma^{2,3}. We used 30 cm H₂O because this compressive dose has been previously established to trigger pathologic remodeling events associated with asthmatic bronchospasm. In our original paper on UJT in airway epithelium⁴ we performed dose-response experiments with 0, 10, 20 and 30 cm H₂O, and found a sharp difference in response between 20 and 30 cm H₂O, but no further increase in response between 30 and 40 cm H₂O. This indicates that the threshold in the dose-response curve lies between 20 and 30 cm H₂O, with the latter being sufficient to elicit the maximal response.

Comment: I fully agree that these two conditions separately trigger the expected phenotypes. However, since the idea of the present paper is to compare them, the “right” conditions are difficult to identify.

Response: That is true, and in this revised manuscript we have tried to define those conditions more clearly. For the first time, we induced and compared these two distinct pathways in one system. We

have revised the manuscript to more clearly define and justify the experimental conditions used to initiate and sustain pEMT or UJT (lines 84-93, 401-412).

Comment: At that point the vertex model-based simulations should be very valuable. And it is in some respect. The observed trends qualitatively reproduce the experimental observations and the interpretation that the authors give although qualitative seem sound and give a very plausible picture of the two transitions. However, it seems to me that it should be possible to go further and to extract real numbers from the simulations. In other words, to somehow “fit” the theoretical behaviors on the actual situation. Why aren't tortuosity, aspect ratio or pack sizes directly compared between the experiments and the theory? For instance, the pack size encompasses typically 10 cells in the theory and is much larger in the UJT experiments.

Response: We thank the reviewer for this comment. In response, we have now extended our analysis in that direction. We have taken great care with the choice of indicators to compare directly against experimental results. For example, the aspect ratio (AR) and shape index (q) are both dimensionless measures of cell shape morphologies that are also relatively insensitive to imaging resolution limits and segmentation errors. We therefore compare the AR and q directly between experimental and simulation results in Fig. 4g and show that the simulation paths (changing p_0 vs changing v_0) correspond very well to the experimental results for pEMT and UJT. These simple metrics clearly demonstrate the distinct nature of the two transition paths.

For the tortuosity, we have now added a graph to Extended Data Figure 6 (panel d) showing the direct comparison of AR versus edge tortuosity from the model and experimental data. This additional figure shows that again the pEMT experiment data and simulation data corresponding to varying v_0 agree fairly well.

Finally, we appreciate and have taken the reviewer's suggestion to directly compare pack size between experiment and theory and have added new data to the manuscript (Figure 4h). In the model, speed is not in physical units, and pack size depends strongly on the size of the system. Therefore, we do not directly compare the raw values for speed and pack size between experiment and theory. Instead, each of these are scaled by a global maximum, and we directly compare scaled speed and scaled pack size between experiment and theory. We find a strong agreement for cells undergoing UJT, supporting our conclusion that in both the model and in the data, collective movement allows cells to move cooperatively and quickly. Thank you for this suggestion.

Comment: The introduction of the persistence time and its importance would also deserve more explanation. As it is, it appears unrelated to the other parameters and although somehow natural, it is unclear what are the conclusions derived from Figure 4f. In general, the authors should include more elements of the model in the main text.

Response: We agree that more information about persistence time and its importance are needed. To extend the conclusions derived from Figure 4f we now have added Figure 4h, which directly compares experiment and theory to show the positive relationship between pack size and migration speed (lines 277-286, 305-308).

Comment: In the last part, the authors mention the processes of invasion through adjacent tissues. This is quite different from the situation studied here and the phrasing should be revised.

Response: We agree and have altered the text accordingly (lines 337-339).

Comment: Finally, I don't understand the last part mentioning the potential role of UJT in the first steps of pEMT. I don't see to what the authors refer to (next to last paragraph).

Response: Using a variety of markers, UJT and EMT are seen to exhibit similar dynamic and structural signatures at early time points after stimulation but exhibit divergent signatures at later time points (Fig. 1, 24 hrs versus 48-72 hrs). These data indicate that as cells undergo pEMT they might first unjam; prior to loss of epithelial layer integrity and prior to strong expression of mesenchymal markers, cells may unjam by elongating and beginning to rearrange with their neighbors. This hypothesis is consistent with our data and with the report by Atia, et al⁵ as discussed in the text. To clarify the point, we added references to the relevant figures (lines 354-355).

Comment: All in all, the article is beautifully written, it addresses an important point, and it provides some clear answers, even if the above-mentioned points question the black-and-white conclusions. I would therefore favor its publication in Nature Communications providing that the comments above are addressed in the text to make sure the reader can understand the hypotheses and limitations of the analysis.

Response: We appreciate this comment. We revised the text to address the comments raised by the reviewer as described above.

Reviewer #3 (Remarks to the Author):

Comment: The paper entitled “In airway epithelial cells, the unjamming transition is distinct from the epithelial-to-mesenchymal transition” from Mitchel et al., presents original experimental results related to the transition from stable epithelia to dynamic structures where the cellular organization can remodel. The authors use two different types of perturbation, mechanical pressure and exposure to TGF-beta1, to trigger this transition. Interestingly they found that, although both approaches enable cell migration, the qualitative behavior of the cells was different in these two situations. Using a model, the authors interpret the role of TGF-beta1 as reducing junctional tension and allowing the cells to explore a richer landscape of shapes, facilitating rearrangements. Pressure, however, was seen to trigger significant motion and was interpreted in the model as increased cellular propulsion.

Demonstrating both patterns of behavior [UJT and pEMT] with the same experimental system (but using different treatments) is interesting and gives a lot of value to his paper. It brings together different interpretations of the transition from stable epithelia to motile layers. As such, the experimental content of paper is worth publishing, and the model provides useful ideas to interpret the data. There are various aspects that the authors may need to clarify before the paper can be published.

Response: Thank you.

Comment: 1. The compression protocol needs to be explained more thoroughly. What is being compressed and how? Is it uniaxial compression? What pressure is imposed?

Response: Reviewer 2 posed a similar question. We had failed to make it clear in this original submission but have now included Extended data Figure 1 to explain the set-up and have revised the methods section (lines 406-412).

As now described in the text (lines 84-93), we used apico-to-basal compression (not uniaxial compression) not only because this mechanical stimulus is transduced by the cells but also because it mimics the manner in which asthmatic bronchospasm compresses the airway epithelium and thus promotes airway remodeling^{3 2}. We used 30 cm H₂O because this compressive dose has been previously established to trigger pathologic airway remodeling that is a hallmark of asthma. In our original paper on UJT in airway epithelium⁴ we performed dose-response experiments with 0, 10, 20 and 30 cm H₂O, and found a sharp difference in response between 20 and 30 cm H₂O, but no further increase in response between 30 and 40 cm H₂O. This indicates that the threshold in the dose-response curve lies between 20 and 30 cm H₂O, with the latter being sufficient to elicit the maximal response.

Using this *in vitro* system, work by our group and others has established that mechanical compression induces events that occur in the remodeled asthmatic airway. These events include increased matrix deposition, goblet cell hyperplasia, and airway smooth muscle hyperplasia and hypercontraction, as well as production of asthma-associated mediators including YKL-40 and tissue factor-containing exosomes^{3, 6-13}. All of these pathophysiological events are triggered by mechanical compression in the absence of any inflammatory cells. Our *in vitro* work was validated in humans by Grainge et al¹⁴. Together, this body of literature has established *in vitro* and in mild asthmatic patients *in vivo* that even in the absence of inflammatory cells, the mechanical effects of bronchoconstriction are sufficient to drive aberrant airway remodeling that is a cardinal feature of asthma^{7, 12, 13}.

Comment: How could cells sense this mechanical signal? Why would they move under compression? Apologies if I missed this explanation in the submission.

Response: Great questions. This question is the focus of our recently published work using RNA seq, which has identified pathways altered by mechanical compression¹⁵. Our earlier work has also shown that a variety of intracellular signaling pathways are involved including EGFR, ERK, and PKC pathways, and this work shows that ERK signaling is required for compression-induced UJT (Extended data Fig 5)^{3, 11, 16}. Molecular mechanism of compression-induced unjamming continues to be a major focus of our ongoing research.

Comment: 2. The vertex model extension used in this study illustrates well the point being made. But this model is an incremental improvement on the work of many others over the past 10 years. I'm not saying this as a negative point, but more credit needs to be given to precursors of the ideas developed here, highlighting the novelty where appropriate. The authors discuss in particular on page 8 (bottom) the benefits on their approach based on a persistence time. It should be clear that a number of older studies from the early 2010s have already studied the role of migratory forces as a trigger for a dynamic transition, in some cases in a way that is very similar to the present model. See for instance doi:10.1098/rsif.2012.0448 as well as its extension to the case of heterogeneous cells populations, doi:10.1098/rsos.161007.

Response: Thank you for catching this omission. We have now added text (lines 277-279) and appropriate references to emphasize these important points.

Comment: Along the same line, when the authors say in their conclusion that "mixed epithelial and mesenchymal characteristics, and the interactions between them, are thought to be essential for carcinoma cell invasion and dissemination, but how UJT might fit into this physical picture remains

unclear”, it is worth pointing that such questions have already been touched upon by people studying collective movements. Although they did not call their transition “UJT”, the underlying models would have very similar physical foundations.

Response: Again, thank you for pointing out this omission. In this revision we have added citations highlighting experimental and modeling work on mixed epithelial and mesenchymal characteristics, along with work hinting that the epithelial-mesenchymal spectrum is insufficient to account for collective motion in all circumstances (lines 364-366).

1. Yang, J. *et al.* Guidelines and definitions for research on epithelial-mesenchymal transition. *Nat Rev Mol Cell Biol* (2020).
2. Grainge, C.L. *et al.* Effect of bronchoconstriction on airway remodeling in asthma. *N Engl J Med* **364**, 2006-2015 (2011).
3. Tschumperlin, D.J. *et al.* Mechanotransduction through growth-factor shedding into the extracellular space. *Nature* **429**, 83-86 (2004).
4. Park, J.A. *et al.* Unjamming and cell shape in the asthmatic airway epithelium. *Nat Mater* **14**, 1040-1048 (2015).
5. Atia, L. *et al.* Geometric constraints during epithelial jamming. *Nature Physics* **14**, 613-620 (2018).
6. Park, J.A., Fredberg, J.J. & Drazen, J.M. Putting the Squeeze on Airway Epithelia. *Physiology (Bethesda)* **30**, 293-303 (2015).
7. Swartz, M.A., Tschumperlin, D.J., Kamm, R.D. & Drazen, J.M. Mechanical stress is communicated between different cell types to elicit matrix remodeling. *Proc Natl Acad Sci U S A* **98**, 6180-6185 (2001).
8. Tschumperlin, D.J., Shively, J.D., Kikuchi, T. & Drazen, J.M. Mechanical stress triggers selective release of fibrotic mediators from bronchial epithelium. *Am J Respir Cell Mol Biol* **28**, 142-149 (2003).
9. Mitchel, J.A. *et al.* IL-13 Augments Compressive Stress-Induced Tissue Factor Expression in Human Airway Epithelial Cells. *Am J Respir Cell Mol Biol* **54**, 524-531 (2016).
10. Park, J.A., Drazen, J.M. & Tschumperlin, D.J. The chitinase-like protein YKL-40 is secreted by airway epithelial cells at base line and in response to compressive mechanical stress. *J Biol Chem* **285**, 29817-29825 (2010).
11. Park, J.A. *et al.* Tissue factor-bearing exosome secretion from human mechanically stimulated bronchial epithelial cells in vitro and in vivo. *J Allergy Clin Immunol* **130**, 1375-1383 (2012).
12. Park, J.A. & Tschumperlin, D.J. Chronic intermittent mechanical stress increases MUC5AC protein expression. *Am J Respir Cell Mol Biol* **41**, 459-466 (2009).
13. Lan, B. *et al.* Airway epithelial compression promotes airway smooth muscle proliferation and contraction. *Am J Physiol Lung Cell Mol Physiol* (2018).
14. Grainge, C.L. *et al.* Effect of bronchoconstriction on airway remodeling in asthma. *N Engl J Med* **364**, 2006-2015 (2011).
15. Kilic, A. *et al.* Mechanical forces induce an asthma gene signature in healthy airway epithelial cells. *Scientific reports* **10**, 966-966 (2020).
16. Tschumperlin, D.J. *et al.* Bronchial epithelial compression regulates MAP kinase signaling and HB-EGF-like growth factor expression. *Am J Physiol Lung Cell Mol Physiol* **282**, L904-911 (2002).

REVIEWER COMMENTS

Reviewer #2 (Remarks to the Author):

The authors have taken into account most of my remarks and I find the new version more balanced. However, I think it would be worthwhile making some of the assumptions clearer in the text. In this line, the arguments of the author's rebuttal should be explicitly in the text (eg last paragraph, p2 of the rebuttal would go p. 3 of the manuscript). Since pEMT is by nature so difficult to define, the choice of markers is critical and the authors should justify their approach. Also, the choice of a pressure of 30 cm H₂O shouldn't be justified by the asthma physiology (that much has been done many times in the authors' previous publications) but with physical arguments since the authors need a meaningful comparison with the other situations. All in all, the paragraph added in the new version p 3 and parts of the new paragraph p 13 should be put together and expended with a discussion over pEMT better definition (or lack of it) p3.

The new panels are nice additions to the manuscript. Fig 4h and sm6 in particular are interesting. Some questions remain however: How do the authors conclude from their analysis of Figure 4h that EMT or jammed cells are not described by the DVM? Why is the shape of the curve speed(I_0) in figure 4f so different from the first version ? I understand Fig 4h as speed vs pack size after normalization with DVM data from Fig 4f. However, the data of Fig 4F don't seem to be consistent with the dotted line of Fig 4h. The authors should double check their analysis or explain better how the DVM curve is constructed in Fig 4h.

In conclusion, this is an interesting and important work and it should be now published after the above comments are addressed.

Reviewer #3 (Remarks to the Author):

The authors have addressed my comments and provided additional information where appropriate. I am very happy to recommend the publication of the paper.

Response to the Editor:

We thank both Reviewers for their efforts in reviewing the revised manuscript. The additional comments from reviewer #2 were immensely helpful and are most appreciated. Here we addressed the remaining concerns from reviewer #2 and we believe this has improved the clarity and coherence of the manuscript. Substantive changes from the previous draft are shown in blue.

Reviewer #2 (Remarks to the Author):

Comment: The authors have taken into account most of my remarks and I find the new version more balanced. However, I think it would be worthwhile making some of the assumptions clearer in the text.

Response: Thank you. We have revised the text as suggested, described below, to clarify our assumptions.

Comment: In this line, the arguments of the author's rebuttal should be explicitly in the text (eg last paragraph, p2 of the rebuttal would go p. 3 of the manuscript).

Response: We have now explicitly explained our reasoning for the use of the dose of 30 cm H₂O in the revised text (lines 94-97), as originally outlined in the rebuttal letter.

Comment: Since pEMT is by nature so difficult to define, the choice of markers is critical and the authors should justify their approach.

Response: We agree. Therefore, in addition to citing the recent consensus report on characterizing pEMT/EMT, we have explicitly stated the details of our approach to characterize the relative epithelial or mesenchymal state of the cells in the main text (lines 102-111).

Comment: Also, the choice of a pressure of 30 cm H₂O shouldn't be justified by the asthma physiology (that much has been done many times in the authors' previous publications) but with physical arguments since the authors need a meaningful comparison with the other situations.

Response: We appreciate this comment and we agree. We have therefore expanded the text to more easily allow for comparisons with other studies and situations (lines 83-94). We emphasize that (1) this level of compression mimics that experienced in vivo during bronchoconstriction, as determined by both modeling and experiment studies; (2) this level of compressive stress is comparable to that used in the literature and known to trigger intracellular signaling; and (3) this dose has been shown to trigger both pathologic remodeling events

associated with asthma and a robust unjamming transition. We have further explicitly stated that that our previous work shows a dose-response for increasing levels of compression from 0 to 30 cm H₂O, as originally described in our rebuttal letter and as requested to be moved into the main text.

Comment: All in all, the paragraph added in the new version p 3 and parts of the new paragraph p 13 should be put together and expanded with a discussion over pEMT better definition (or lack of it) p3.

Response: Thank you. In taking into account the specific suggestions above, we have expanded our explanations of how we triggered and evaluated both UJT and pEMT, as described above.

Comment: The new panels are nice additions to the manuscript. Fig 4h and sm6 in particular are interesting.

Response: Thank you.

Comment: Some questions remain however: How do the authors conclude from their analysis of Figure 4h that EMT or jammed cells are not described by the DVM?

Response: Thank you for this comment. We now realize that the caption for Fig. 4 generated this confusion and have clarified the text.

We did not mean to imply that pEMT or the jammed state are not described by the DVM. By contrast, we wish to emphasize that the DVM in fact captures the structural and dynamic characteristics of all the states explored in our experiments as a function of distinct parameter regimes. As such, the broad goal of Fig. 4 is to show that different regimes of the DVM in fact explain the distinct characteristics of pEMT versus UJT.

First, we show that jammed cells are described by states characterized by small values of preferred shape index p_0 at low enough single-cell motility v_0 . These states are shown in Fig. 4a-(i) and Fig. 4c-(iv), respectively. Next, we demonstrate that, when single cell motility is kept small, as shown in Fig. 4 a, b, then increasing p_0 captures the single-cell shape characteristics of pEMT. The agreement between DVM in this regime and experimental data for cells undergoing pEMT is clear from Fig. 4g. We further show that when single-cell motility v_0 is increased, as shown in Fig. 4c, d, then the single-cell shape characteristics of UJT are captured

by the DVM. Again, the agreement between DVM in this regime and experimental data for cells undergoing UJT is clear from Fig. 4g.

Last, we looked at the effects of changing the single cell persistence length l_0 (Fig. 4e, f). Here, instead of observing single-cell shape characteristics, we instead observe collective dynamics, and find the existence of cooperative dynamic packs, as shown in our experimental data. Again, we show that in this regime, we can capture an aspect of the experimental observations from UJT. Building on all of this, in Fig. 4h we compare the normalized average cell speeds and normalized median dynamic pack sizes. In the regime of sufficiently large single cell motility v_0 , and when all other simulation parameters are kept fixed, dynamic pack sizes are increased in tandem with increasing the single cell persistence length l_0 . This behavior is consistent with what occurs during UJT, but distinct from pEMT, as during pEMT neither cell migration speeds nor large dynamic packs develop. Therefore, while the DVM as a whole can describe the jammed state as well as the characteristics of cells undergoing pEMT or UJT, in the case of increasing single cell persistence length l_0 , we find a concordance solely between that regime and UJT.

Thus different single cell parameters of the DVM (p_0, v_0, l_0), when varied independently describe different states observed in our experimental systems and together provide a complete physical picture of pEMT and UJT. We have now improved the caption of Fig. 4 to clarify these connections.

Comment: Why is the shape of the curve $\text{speed}(l_0)$ in figure 4f so different from the first version?

Response: Assuming the reviewer meant the behavior of “cell speed” as function of l_0 in Fig. 4f, we agree that its shape is quantitatively different from the first version, but we note no qualitative differences. The origin of the difference is the following: in response to suggestions from both reviewers, we revisited our calculations of dynamic pack sizes from the DVM. We performed a new measurement of the linear pack sizes which required an improved averaging and sampling. The cell speeds were calculated from the simulated trajectories using displacements over a time interval chosen suitably to reduce the contribution from very short-lived dynamic packs, predominantly observed when l_0 is comparable to one cell diameter or less. This method has already been included in the supplementary text on DVM in page 4-5.

Comment: I understand Fig 4h as speed vs pack size after normalization with DVM data from Fig 4f. However, the data of Fig 4F don't seem to be consistent with the dotted line of Fig 4h. The authors should double check their analysis or explain better how the DVM curve is constructed in Fig 4h.

Response: Thanks for pointing this out. You are right that the pack sizes reported in Fig. 4f are not the same as those shown by the dotted line in Fig. 4h. In fact, in this version of Fig. 4f we showed only the behavior of largest dynamic pack sizes as a function of l_0 while in Fig. 4h we compared median dynamic pack sizes across DVM and experiments. This is the source of the discrepancy which you point out.

We note that the caption of Fig. 4h is not very explicit about the above difference from Fig. 4f. To avoid any confusion, we have updated Fig. 4f to show the statistics of median dynamic pack sizes as well. We have rewritten the captions of Fig. 4f and h to reflect this change.

We note that we have chosen to show the statistics of both the median and the largest pack sizes in Fig. 4f. The motivation of showing the largest pack sizes comes from your previous suggestion to compare pack sizes between the theory and experiment. You previously noted that the pack sizes in the DVM are typically much smaller compared to those in the experiments, which we have determined is due to the relatively smaller system size of the simulations compared to the experiments. That is, the largest pack sizes observed in the DVM clearly show that for large enough l_0 the DVM can describe dynamic packs as large as the system size (also illustrated in Fig. 4e-ix). This means if we increase the window of our simulation, we will see packs as big as the ones observed in our experiments.

Comment: In conclusion, this is an interesting and important work and it should be now published after the above comments are addressed.

Response: Thank you for your consistently insightful and helpful suggestions. Your comments have been invaluable in improving our manuscript.

REVIEWERS' COMMENTS

Reviewer #2 (Remarks to the Author):

The authors have followed my suggestions and have made the concepts they rely on more explicit. Readers should now be able to critically read the manuscript. I consider this article as an important study on a crucial problem and I recommend its publication in Nature Communications.